# Proteomic Response of *Paracoccidioides brasiliensis* Exposed to the Antifungal 4-Methoxynaphthalene-N-acylhydrazone Reveals Alteration in Metabolism

**DOI:** 10.3390/jof9010066

**Published:** 2022-12-31

**Authors:** Lívia do Carmo Silva, Kleber Santiago Freitas e Silva, Olívia Basso Rocha, Katheryne Lohany Barros Barbosa, Andrew Matheus Frederico Rozada, Gisele de Freitas Gauze, Célia Maria de Almeida Soares, Maristela Pereira

**Affiliations:** 1Laboratory of Molecular Biology, Institute of Biological Sciences, Federal University of Goiás, Goiás 74690-900, Brazil; 2Department of Chemistry, State University of Maringa, Paraná 87020-900, Brazil

**Keywords:** paracoccidioidomycosis, *N*-acylhydrazone, antifungal, proteome

## Abstract

**Background:** Paracoccidioidomycosis is a neglected mycosis with a high socioeconomic impact that requires long-term treatment with antifungals that have limitations in their use. The development of antifungals targeting essential proteins that are present exclusively in the fungus points to a potentially promising treatment. **Methods:** The inhibitor of the enzyme homoserine dehydrogenase drove the synthesis of *N’*-(2-hydroxybenzylidene)-4-methoxy-1-naphthohydrazide (AOS). This compound was evaluated for its antifungal activity in different species of *Paracoccidioides* and the consequent alteration in the proteomic profile of *Paracoccidioides brasiliensis*. **Results:** The compound showed a minimal inhibitory concentration ranging from 0.75 to 6.9 μM with a fungicidal effect on *Paracoccidioides* spp. and high selectivity index. AOS differentially regulated proteins related to glycolysis, TCA, the glyoxylate cycle, the urea cycle and amino acid metabolism, including homoserine dehydrogenase. In addition, *P. brasiliensis* inhibited protein synthesis and stimulated reactive oxygen species in the presence of AOS. **Conclusions:** AOS is a promising antifungal agent for the treatment of PCM, targeting important metabolic processes of the fungus.

## 1. Introduction

Paracoccidioidomycosis (PCM) is a systemic mycosis prevalent in Latin America, which is associated with lesions in different organs with a multiplicity of clinical conditions. The main PCM case reports are related to specific social segments such as males, rural workers and those associated with land management and aging between 30 and 50 years of age. Clinical manifestations are usually debilitating, preventing individuals affected by the mycosis to perform daily tasks and work activities. The result is a significant socio-economic impact [1]. It is noteworthy that immunocompromised patients develop a more acute clinical form of the disease and show a higher mortality rate [2,3]. 

The treatment of PCM is a challenge. The high antifungal dosage is administered for a long time, ranging from 6 months to 2 years. In addition, the treatment consists of two phases, firstly the attack phase to immediately control the infection, leading to the reduction in the fungal load and the symptoms of the disease. Secondly, the maintenance phase, which focuses on the permanent cure of the patient. Three main therapeutic options have been commonly used, sulfonamides, itraconazole and amphotericin B [1,4]. The targets of these antifungals are restricted to the plasma membrane and folic acid synthesis [5].

Based on the issues related to the current antifungals used against PCM and the possibility of fungal resistance that results from indiscriminate use of antifungals, there is a need to research new therapeutic targets and antifungal agents against this disease [6]. Hence, our group has focused on the search for compounds that act on new therapeutic targets, such as methylcitrate synthase [7], isocitrate lyase [8], malate synthase [9] and homoserine dehydrogenase (HSD) [10]. HSD is a key enzyme of the aspartic acid pathway involved in the biosynthesis of the amino acids threonine, methionine and lysine in fungi and plants. This enzyme converts L-aspartate-4-semialdehyde into L-homoserine and vice versa. HSD is not expressed in humans; thus, it represents a potential therapeutic target and consequently, inhibitor compounds have been tested against this enzyme [11,12].

Bagatin et al. [11] performed a virtual screening and a molecular docking assay and selected four potential inhibitors of HSD for in vitro antifungal assays. Among these compounds, the two most promising showed a minimum inhibitory concentration (MIC) ranging from 32 to 64 μg/mL against *Paracoccidioides* spp.. These inhibitors were posteriorly used as hits for developing new compounds, thus proposing a synthetic series of structures derived from the 4-methoxynaphthoic acid. Their antifungal activity against *P. brasiliensis* were evaluated. Two of them showed promising antifungal activity (MIC of 16 and 64 μg/mL) against fungal cells and the carbohydrazide derivative showed activity against the *P. brasiliensis* and *P. lutzii* species, with MIC values ranging from 8 to 32 μg/mL. In a different approach, *N-*acylhydrazones’ derivatives were synthesized using the carbohydrazide as the lead compound. The activity of the *N-*acylhydrazones against *P. brasiliensis* and *Mycobacterium tuberculosis* was evaluated and *N’*-(2-hydroxybenzylidene)-4-methoxy-1-naphthohydrazide (AOS) showed a very promising activity against these fungal species, with a MIC value of 0.5 µg/mL [13].

Given the pharmacological potential of *N*-acylhydrazones [14], we investigated the antifungal activity of the compound AOS in different *Paracoccidioides* species, as well as the metabolic alterations caused by the compound in *P. brasiliensis* through a proteomic approach.

## 2. Materials and Methods

### 2.1. Synthesis of AOS

AOS was synthetized according to Rozada et al. [13]. Initially, 4-methoxynaphthoic acid was used as the precursor and it was converted into ethyl 4-methoxy-1-naphthoate through the reaction with thionyl chloride (1.2 eq) and ethanol reflux for 16 h. Further, the ethyl ester was converted into 4-methoxy-1-naphthohydrazide by reflux with excess (50 eq) of hydrazine monohydrate (NH_2_NH_2_.H_2_O 80%) for 24 h. Then, the carbohydrazide derivative (1.0 eq) was reacted with salicylic aldehyde (1.0 eq), using acid catalysis of HCl in DMSO for 1 h at room temperature to obtain AOS.

### 2.2. Cultivation Conditions of Paracoccidioides spp.

*P. lutzii* (*Pb*01), *P. brasiliensis* (*Pb*18), *P. americana* (*Pb*03) and *P. restrepiensis* (*EPM*83) were cultivated in Fava-Netto medium (1% peptone, 1% brain heart infusion, 0.5% meat extract, 0.5% yeast extract, 0.3% protease peptone, 4% glucose, 0.5% NaCl) and 5 μg/mL gentamycin for 72 h at 37 °C under shaking. The inhibitory and fungicidal concentration, temporal cell viability, proteome, protein dosage, enzyme activity inhibition assay and quantification assay of reactive oxygen species were performed in the chemically defined culture medium RPMI-1640 (Sigma-Aldrich, Missouri, EUA). The growth assay with methionine supplementation was performed in McVeigh/Morton as described by Restrepo and Jiménez, 1980 [15].

### 2.3. Determination of Inhibitory and Fungicidal Concentration

Minimum Inhibitory Concentration (MIC) was determined using the microdilution technique as described by Silva et al. [16]. AOS was serially diluted (125 to 0.06 µg/mL) in RPMI-1640 medium and incubated with 1×10^5^ cells/mL at 37 °C under agitation for 48 h. Subsequently, 20 μL of 0.02% resazurin solution was added and the microplate was incubated for 24 h at 37 °C. The MIC value was determined visually based on the resazurin conversion (blue) into resorufin (pink). Cells grown in the absence of AOS comprised the positive control group. The negative control comprised samples without fungal cells for all the concentrations of the compound that were tested. To determine the Minimum Fungicidal Concentration (MFC), 20 μL of MIC and positive control samples were added to solid Fava-Netto medium. The plates were incubated at 37 °C for seven days and MFC was defined as the lowest concentration where fungus growth was not visualized.

### 2.4. Cytotoxicity Assay

Cytotoxicity was evaluated using BALB/c 3T3 clone A31 cells (ATCC^®^ CCL-163™) according to Silva et al. [16]. The cells were cultured in DMEM medium (Sigma-Aldrich) supplemented with 10% fetal bovine serum. Then, 1×10^5^ cells/mL were incubated in different concentrations of AOS (1500 to 0.48 μg/mL) at 37 °C for 48 h with 5% CO_2_. Following that, 20 μL of 0.02% resazurin was added to each well and incubated for 24 h. Cytotoxicity was determined visually based on the dye reduction and resazurin color change, where blue indicated growth inhibition and pink meant cell viability. The calculation of the selectivity index was defined as the ratio between the cytotoxic concentration and the inhibitory concentration of AOS.

### 2.5. Temporal Cell Viability

Samples containing 1 × 10^5^ cells/mL of *P. brasiliensis* in the presence of 0.75 μM (0.48 μg/mL) of AOS and in the absence of the compound were incubated at 37 °C under agitation. Subsequently, aliquots of 1 mL were removed after 3, 6, 9, 12, 24, 48 and 72 h of incubation. The viability of cells in the samples was determined by counting cells treated with trypan blue dye 0.4% in a Neubauer chamber.

### 2.6. Proteome of P. brasiliensis in the Presence of AOS

A sample comprising 1 × 10^5^ cells/mL and incubated in the presence of 0.75 μM (0.48 μg/mL) of AOS (treatment condition) or in the absence of the compound (control condition) at 37 °C for 12 h was centrifuged for 10 min at 3000 g, 4 °C. The supernatant was discarded and the fungal cell concentrate was resuspended in ammonium bicarbonate buffer (NH_4_HCO_3_) at 50 mM, pH 8.5 and protein extraction was performed by cell lysis using glass beads as described by Rocha et al. [17]. The proteins were digested according to Murad et al. [18]. Briefly, RapiGEST^TM^ 0.2% (*v*/*v*) (Waters Corporation) was added to samples containing 150 µg of the protein extract following incubation at 80 °C for 15 min. Subsequently, 7.5 µL of 100 mM dithiothreitol (DTT) (GE Healthcare, IL, USA) was added and samples were incubated at 60 °C for 30 min. Then, 7.5 μL of 300 mM iodoacetamide solution (GE Healthcare) was added and samples were again incubated at room temperature for 30 min and protected from light. Eventually, 30 µL of trypsin (50 ng/µL) was added and the samples were digested at 37 °C for 16 h. Precipitation of RapiGEST^TM^ was performed by adding 10 μL of 5% (*v*/*v*) trifluoroacetic acid (Sigma-Aldrich), followed by incubation at 37 °C for 90 min. The sample was centrifuged at 13,000× *g* for 30 min, the supernatant lyophilized using a vacuum concentrator and the digested samples were stored at −20 °C until further analysis by mass spectrometry.

The digested peptides were resuspended in 45 µL of a solution containing 20 mM ammonium formate and 150 fmol/µL of Rabbit Phosphorylase B (Waters Corporation, MA, USA). The separation of tryptic peptides by nanoscale liquid chromatography was performed using a nanoACQUITY^TM^ system (Waters Corporation). Peptides were separated using a five-fraction gradient of 10.8%, 14%, 16.7%, 20.4% and 65% acetonitrile and 0.1% (*v*/*v*) formic acid at a rate flow of 2000 µl/min. The source was operated in nano-ESI (+) positive ionization mode. To perform an external calibration, the masses were corrected based on Glu-fibrinopeptide B (Sigma Aldrich) of molecular mass 785.8486. A GFP solution of 50% (*v*/*v*) methanol and 0.1% (*v*/*v*) formic acid at the final concentration of 200 fmol/µL, supplied by the mass spectrometer’s NanoLockSpray source was used. The mass spectrometry analysis was performed on a Synapt G1 MS^TM^ (Waters) equipped with a NanoElectroSpray source and two mass analyzers: a quadrupole and a time of flight (TOF) operating in V mode. Data were obtained with the instrument in the MS^E^ mode. Samples were analyzed in three replicates. Raw MS data were processed using ProteinLynx Global Server version 2.4 (Waters Corporation). Data were subjected to automatic subtraction of background, isotopy and convolution from the state of charge. After processing, each ion comprised an exact mass retention time that contained retention time, intensity-weighted average charge, lower molecular weight based on charge and *m*/*z*. Then, the processed spectra were searched against the *Pb*18 protein sequences (https://www.uniprot.org/proteomes/UP000001628, accessed on 12 July 2021) along with the reverse sequences. Differentially expressed proteins were categorized according to the FungiFun2 database available at https://elbe.hki-jena.de/fungifun/ (accessed on 12 July 2021). The submission of the proteomics’ results was assigned the identifier PASS03787 in the PeptideAtlas repository.

### 2.7. Protein Dosage

Fungal cells were cultured in the presence and absence of AOS for 12 h. Then, cells were centrifuged at 3000 g and the supernatant discarded. Next, 1 mL of ammonium bicarbonate buffer (50 mM, pH 8.8) was added to 0.5 g of pellet. Cytoplasmic protein extraction was performed by cell lysis using glass beads and a bead-beater shaker for five cycles of 30 s, with intervals of 30 s on ice. The lysate was centrifuged for 20 min at 5000 g and the supernatant was collected. The protein quantification was performed using Bradford reagent (Sigma-Aldrich) as described by Bradford [19]. Statistical analysis was performed using the Student’s *t*-test and values of *p* ≤ 0.05 were considered statistically significant.

### 2.8. Growth Assay with Methionine Supplementation

Samples with 1 × 10^5^ cells were cultured in McVeigh/Morton medium supplemented or not with 10 mM methionine in the presence or absence of 0.75 μM by 12 h. The growth was verified by reading the absorbance at 600 nm. The statistical difference was assessed using the Student’s *t*-test and values of *p* ≤ 0.05 were considered statistically significant.

### 2.9. Enzyme Activity Inhibition Assay

The enzymatic activity of formamidase was verified by measuring ammonia formation as described by Skouloubris et al. [20]. A sample containing 1 µg of protein extract was added to 200 µL of a 100 mM formamide solution, 100 mM phosphate buffer and 10 mM EDTA, pH 7.4. This sample was subsequently incubated for 30 min at 37 °C. After incubation, 400 µL of phenol-nitroprusside and 400 µL of alkaline hypochlorite were added and the samples were incubated again for 6 min at 50 °C. Absorbance was verified at 625 nm. A standard curve was performed to determine the amount of ammonia released. The enzyme activity unit was defined as the amount of formamidase necessary to hydrolyze 1 µmol of formamide that generates one µmol of ammonia.

The enzymatic activity of urease was verified by measuring the formation of ammonium from urea as previously described by Lerm et al. [21]. A sample of 1 mL of buffer (50 mM HEPES and 100 mM urea, pH 7.4) was added to 100 µg of fungal protein extract and incubated at 37 °C for 60 min. Then, 50 µL of this solution were collected and added to 500 µL of phenol (10 g/L) and sodium nitroprusside (50 mg/L). Next, 500 µL of sodium hydroxide (5 g/L) and sodium hypochlorite (8.4 ml/L) solution was added. The sample was incubated at 37 °C for 30 min and the absorbance read at 625 nm. A standard curve was performed to determine the amount of ammonia released. The enzyme activity unit was defined as the amount of urease necessary to hydrolyze 1 µmol of urea that generates 1 µmol of ammonia. 

The statistical comparison between treated and control samples was performed using the Student’s *t*-test, where *p* ≤ 0.05 values were considered differentially significant.

### 2.10. Quantification Assay of Reactive Oxygen Species

The quantification of reactive oxygen species (ROS) was performed by fluorescence microscopy following the protocol established by Rocha et al. [22]. Samples containing 1×10^5^ fungal cells/mL were exposed to 0.75 μM (0.48 μg/mL) of AOS or without the compound for 12 h. Then, 1 mL of each condition was collected and centrifuged at 3000× *g*. The pellet was resuspended in 1 mL of 1X PBS and 25 μM of dichlorodihydrofluorescein 2ʹ, 7ʹ-diacetate dye (DCFH-DC) (Sigma-Aldrich) was added to the samples, which were incubated in the dark for 30 min. The samples were observed under a fluorescence microscope using a 490 – 516 nm wavelength. The quantification of fluorescence intensity was performed using AxioVision V 4.8.2.0. The fluorescence intensity was calculated by the ratio between the logarithmic values of the cell density of the control and treated samples in each time interval. Statistical analysis between control and treated samples was evaluated using Student’s *t*-test and *p* ≤ 0.05 values were considered significantly different.

## 3. Results

### 3.1. Antifungal and Cytotoxic Activity of AOS

Antifungal activity against *Paracoccidioides* spp. was experimentally evaluated in this work based on the analysis of MIC and MFC values. The results showed that the AOS exerted an inhibitory effect on fungus growth at concentrations ranging from 0.75 to 6.09 μM (0.24 to 1.95 μg/mL). *Pb*18 and *Pb*03 were the most sensitive species to AOS, with a MIC value of 0.75 μM (0.24 μg/mL). The concentrations that were required to inhibit the growth of *Pb*EPM83 and *Pb*01 were 1.5 μM (0.48 μg/mL) and 6.09 μM (1.95 μg/mL), respectively (Table 1). AOS showed values of MFC equal to MIC values. Cytotoxic concentration (CC) was also evaluated. AOS inhibited BALB/3T3 mammalian cells at a concentration of 4685.88 µM (1500 μg/mL). The selectivity index for *Pb*18 and *Pb*03 was 6247.84; for *Pb*01 the index was 769.43 and for *Pb*EPM83 was 3123.92 (Table 1).

The temporal viability of *P. brasiliensis* in the presence of AOS was analyzed. The results showed that the viability of *P. brasiliensis* cells was time-dependent and reduced to 27.5% after 24 h. Fungal cell viability was 85.3% after 12 h of exposure to the compound, while in the absence of AOS the cell population viability of the control group was 95.9% (Figure 1). According to the cellular viability test, the time-point of 12 h was chosen to perform the proteomic profile of *P. brasiliensis* in response to exposure to the AOS compound.

### 3.2. Proteomic Response of P. brasiliensis to AOS

The proteomic analysis allowed the identification of 318 differentially expressed proteins by *P. brasiliensis* cells after treatment with AOS for 12 h. The fold-change was 1.3, a commonly used parameter setting for comparative proteomics [23,24]. Differentially expressed proteins were classified into 125 upregulated (Appendix A) and 193 downregulated proteins (Appendix A) and then into functional categories (Figure 2). The most representative classes of the upregulated proteins were unclassified proteins (39), followed by metabolism (32), protein synthesis (16) and energy (15) (Figure 2A). The main classes of downregulated proteins were unclassified proteins (53), followed by protein synthesis (52), metabolism (30) and energy (12). 

### 3.3. Exposure to AOS Inhibits Protein Synthesis

*P. brasiliensis* in the presence of AOS downregulated a large number of proteins related with protein synthesis (Appendix A). To assess whether AOS influenced protein synthesis, proteins were quantified in cells exposed or not to the compound. The results indicated that the amount of protein was significantly lower in the samples treated with the compound than those grown without the compound (Figure 3).

### 3.4. Methionine Supplementation Restores Fungal Growth Exposed to AOS

Several amino acid metabolism proteins were found to be regulated in cells that were exposed to the compound. Among them, enzymes related to the methionine synthesis were downregulated (Appendix A), such as 5-methyltetrahydropteroyltriglutamate-homocysteine S-methyltransferase and S-adenosylmethionine synthase. To verify if this repression could affect the cells’ growth, we supplemented the medium with 10 mM of methionine in order to analyze if the growth would be recovered. In the absence of AOS, the cells maintained their growth in the medium with and without supplementation. However, cells exposed for 12 h to AOS recovered their growth only when the culture medium was supplemented with methionine (Figure 4).

### 3.5. Exposure to AOS Induces Nitrogen Metabolism

Enzymes involved in nitrogen metabolism as formamidase, urease and uricase were upregulated on *P. brasiliensis* after exposure to AOS. Thus, the enzymatic activity of formamidase and urease was performed to verify if the differential expression of these enzymes was correlated with the enzymatic activity. Both urease and formamidase showed significantly higher activity in the group of samples exposed to AOS for 12 h compared to the control group (Figure 5).

### 3.6. AOS Causes Oxidative Stress in Cells

Oxidative stress response enzymes were observed to be upregulated during exposure to AOS, among them thioredoxin reductase and superoxide dismutase. This response is due to the increase in ROS inside the cells. To confirm this hypothesis, we verified whether ROS production was increased in cells exposed to AOS. The fluorescence microscopy showed a significant increase in fluorescence in the cells treated with the compound compared to the control group, indicating an increase in ROS concentration (Figure 6).

## 4. Discussion

Fungal infections are highly prevalent worldwide. The lack of potent new antifungals and the high toxicity of the compounds may contribute to the development of such mycosis. Recently, synthetic and natural compounds with antifungal properties have been extensively investigated due to their biotechnological potential against pathogenic fungal species [25]. Our research group has strived to identify promising antifungal compounds against *P. brasiliensis* in order to establish different therapeutic approaches against PCM. Proposing brand new antifungal drugs is a laborious process mainly due to the similarities between fungal and human cells but biotechnological strategies might improve the quality of life of patients.

HSD is a key enzyme of the aspartic acid pathway involved in the biosynthesis of threonine, methionine and lysine in fungi and plants. It reversibly converts L-aspartate-4-semialdehyde to L- homoserine and is not expressed in human cells. Hence, it is a good therapeutic target. An amino acid deprivation approach in *C. albicans* along with silencing of the precursor gene of HSD showed the importance of this enzyme to cell growth [26]. In addition, a specific HSD inhibitor was effective as an antifungal and its mode of action showed the inhibition of protein synthesis, significantly decreasing the level of amino acids of the aspartate family such as threonine, methionine and isoleucine, which are amino acids intrinsically related to HSD [27]. A virtual screening and molecular docking assay selected potential HSD inhibitors and the most promising compound showed a MIC value of 32 μg/mL [11]. The AOS compound tested herein is a derivative of previous compounds reported as HSD inhibitors [10] and presented antifungal activity against several isolates of *Paracoccidioides*. Our group has used different approaches in the search for antifungal compounds. A virtual screening methodology has identified antifungal active compounds against specific targets of *P. brasiliensis*, such as methylcitrate synthase (ZINC08964784 with MIC value of 6.04 μM) [7] and isocitrate lyase (compound 04,559,339 with MIC of 38 μM) [8]. Considering these results, the AOS compound showed an impressive MIC value against *Paracoccidioides* growth and attention should be paid to this molecule. The proteomic approach was used to elucidate the possible mode of action of several compounds with antifungal activity against *P. brasiliensis*, as argentilactone [28], camphene thiosemicarbazide [29] and curcumin [17]. Here, we also employed this methodology considering that one of the ways in which cells adjust to environmental changes is through altering the pattern of protein expression. The analysis of differentially expressed proteins led us to perform in vitro experimental validations in order to verify the metabolic processes that were altered by AOS.

The total protein quantification showed a lower number of proteins in samples treated with AOS than those grown in the absence of the compound, evidencing a strong ability of the compound to reduce protein synthesis in *P. brasiliensis*. Over the years, several prototypes of antifungal compounds showed the ability to inhibit protein synthesis. Aspirochlorine, for example, is an inhibitor of protein synthesis in *C. albicans* leading to growth impairment. Interestingly, this compound does not have an effect on protein synthesis of mammalian cells [30]. Screening assays pointed to the efficacy of antifungal compounds that are able to inhibit protein synthesis. The compound NSC319726, a thiosemicarbazone, was screened for antifungal properties from the database available on the Developmental Therapeutics Program of NIH/NCI. It inhibited ribosomal biogenesis in *Candida* species, *Aspergillus fumigatus* and *Cryptococcus neoformans* [31]. Proteomic approaches showed that camphene-thiosemicarbazide [29], curcumin [22] and argentilactone [28] also reduced the protein synthesis of *Paracoccidioides* spp.

Human pathogenic fungi express all proteinogenic amino acids [32] and antifungal compounds that inhibit enzymes of amino acid biosynthetic pathways show promising antifungal activity [33,34]. The virtual screening approach showed that inhibiting such enzymes in *Aspergillus fumigatus* leads to metabolic impairment that causes damage to ATP production, cell growth and virulence [34]. Here, S-adenosylmethionine synthase is downregulated in *P. brasiliensis* cells treated with AOS. This enzyme catalyzes the reaction of methionine and ATP and the product of this reaction (S-adenosyl-L-methionine) is the universal methylating agent that regulates metabolites, phospholipids, DNA, ATP levels and several proteins [35]. S-adenosylmethionine synthase is a moonlight protein and a virulent factor in pathogenic fungi [36]. It was differentially expressed in *P. lutzii* cells grown on medium containing propionate as carbon source [37], in the presence of argentilactone [28] and during copper overload [38]. Thus, S-adenosylmethionine synthase is a promising antifungal target against *P. brasiliensis* as it is in other pathogenic microorganisms [39].

Infection onset and progression depend, among other things, on nitrogen metabolism, which affects the regulation of virulence factors and phenotypic dimorphism [40]. Here, proteins involved in nitrogen metabolism were upregulated, such as formamidase. The two substrates of this enzyme are formamide and water and the products are formate and NH_3_. This enzyme belongs to the family of hydrolases, acting on carbon–nitrogen bonds in order to regulate fungal nitrogen metabolism. Conditions that offer nitrogen limitation to pathogens, a very common environmental stress, affect the virulence and antifungal susceptibility [41]. *P. brasiliensis* cells infect macrophages where carbon and nitrogen limitation are prevalent [42]. Interestingly, nitrogen availability influences the drug sensibility of pathogenic fungi, such as *Cryptococcus neoformans*, which showed high survivability and higher tolerance to amphotericin B on nitrogen cultures, as a way of adapting to this stress condition. It has been shown that antifungal compounds that interfere with nitrogen metabolism change the expression of enzymes from such pathways. Urease and uricase were differentially expressed in a proteomic assay that analyzed the antifungal effect of canthin-6-one on *Fusarium oxysporum*, changing nitrogen physiological processes of fungi and leading to the death of pathogen cells [43]. Additionally, the dynamic of the metabolism for pathogen adaptation to host-imposed stresses, such as nitrogen starvation, has been the target for the development of novel antifungals [44].

Oxidative stress response enzymes were upregulated during exposure to AOS, mainly thioredoxin reductase and superoxide dismutase. The antifungal effects of natural or synthetic compounds have been tested in the *Paracoccidioides* genus and most of these compounds cause ROS metabolism alteration in this fungus. Argentilactone led to a differential expression of thioredoxin and superoxide dismutase, the former was downregulated while the latter was upregulated [28]. The proteomic profile of camphene-thiosemicarbazide resulted in the upregulation of thioredoxin [29] and curcumin resulted in the downregulation of superoxide dismutase [17]. This corroborates the hypothesis that such compounds act by increasing the ROS levels that should be dealt with by pathogenic cells, reducing their survivability once all of them, including AOS, impaired fungal growth.

Finally, the overall AOS mode of action points to the production of pyruvate in order to feed the tricarboxylic acid cycle (TCA) and consequently to restore the energy production (Figure 7). All three points of glycolysis regulation showed increased expression, hexokinase, fructose 1,6-bisphosphatase and pyruvate kinase, channeling the substrate towards pyruvate formation. Pyruvate is then converted into acetyl-CoA by pyruvate dehydrogenase, enabling TCA to run and feed the electron transport chain. Interestingly, the main point of TCA regulation, isocitrate dehydrogenase, showed increased expression. Electrons are transferred into the chain but ATP generation might be impaired since proteins from the electron transport chain (cytochrome c oxidase) and ATP synthase are downregulated. To reinforce the idea that cells are trying to commit pyruvate into ATP production, enzymes that requires pyruvate, as substrates are downregulated, and those that release it after the reaction they catabolize are upregulated. This can be observed for proteins related to amino acid metabolism (Figure 7). 

The downregulated enzymes 3-isopropylmalate dehydratase, homocitrate synthase and spermidine synthase exert roles in the amino acid biosynthesis pathway and require pyruvate as substrate. On the other hand, the upregulated 2–amino methyltransferase, cysteine dioxygenase and acylpyruvate hydrolase participate in amino acid degradation pathways and release pyruvate. This is a strong indication that cells are driving pyruvate into acetyl-CoA formation and feeding TCA towards ATP production. Similarly, enzymes that require acetyl-CoA (3-hydroxy-3-methylglutaryl coenzyme A synthase, 5-methyl-tetrahydropteroyltriglutamate homocysteine S-methyltransferase and arginosuccinate synthase) as substrate are downregulated and hydroxymethylglutaryl-CoA lyase that releases acetyl-CoA is upregulated.

## 5. Conclusion

AOS shows promising antifungal activity with a fungicidal effect. The compound differentially regulated proteins of *P. brasiliensis* related to glycolysis, TCA, glyoxylate cycle, the urea cycle and amino acid metabolism, including HSD, a protein absent in humans and whose activity is inhibited by the AOS precursor compound. In addition, AOS induces oxidative stress and decreases protein synthesis.

## Figures and Tables

**Figure 1 jof-09-00066-f001:**
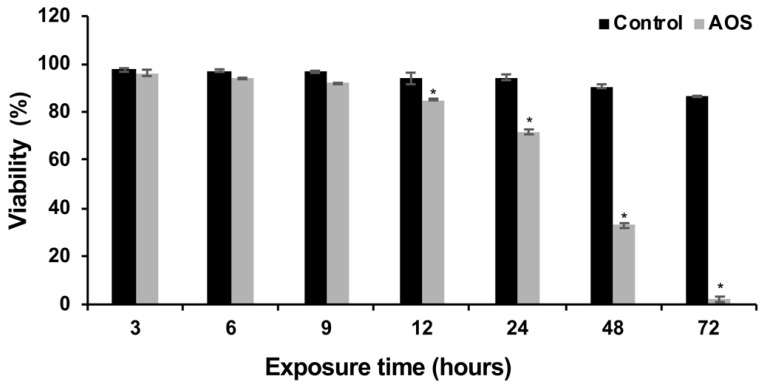
**Cell viability of *P. brasiliensis* in the presence and absence of AOS.** After incubation with the AOS compound, the viability of the cells was verified and compared to the control (cells incubated only with culture medium). * Means significant difference between the samples (*p* ≤ 0.05). Error bars represent the standard deviation of three biological replicates.

**Figure 2 jof-09-00066-f002:**
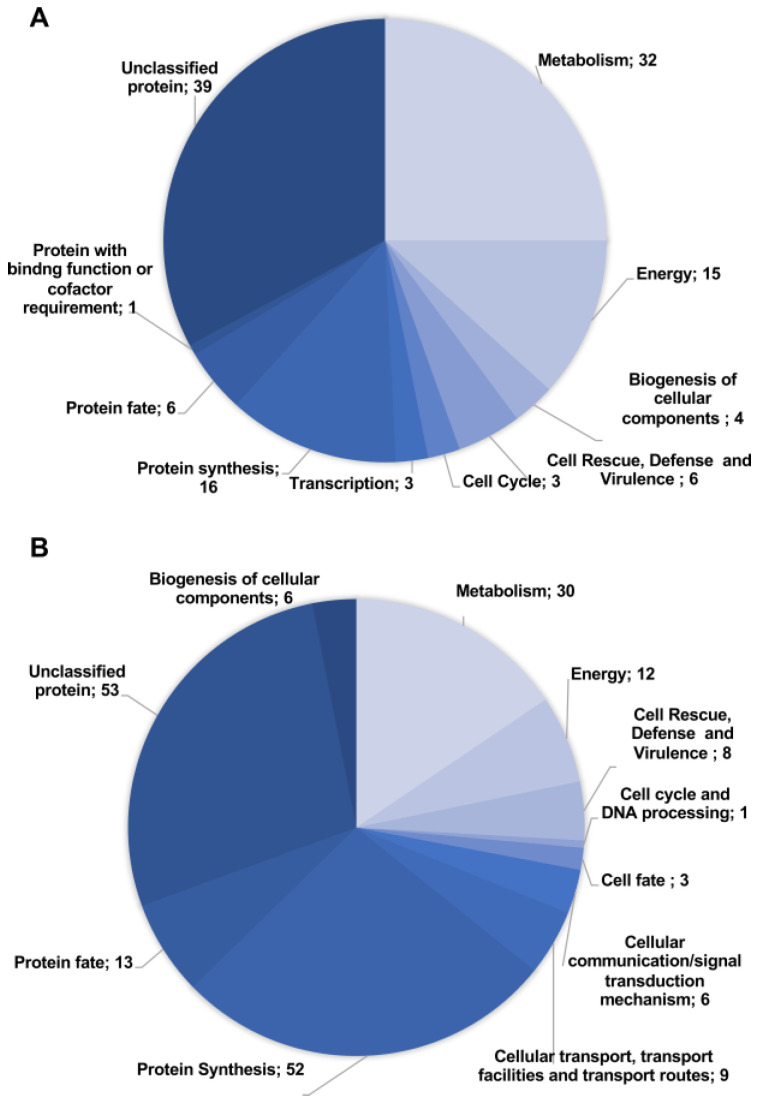
**Representation of the functional classification of differentially regulated proteins by *P. brasiliensis* after exposure to AOS**. (**A**) Proteins with increased abundance, (**B**) Proteins with decreased abundance.

**Figure 3 jof-09-00066-f003:**
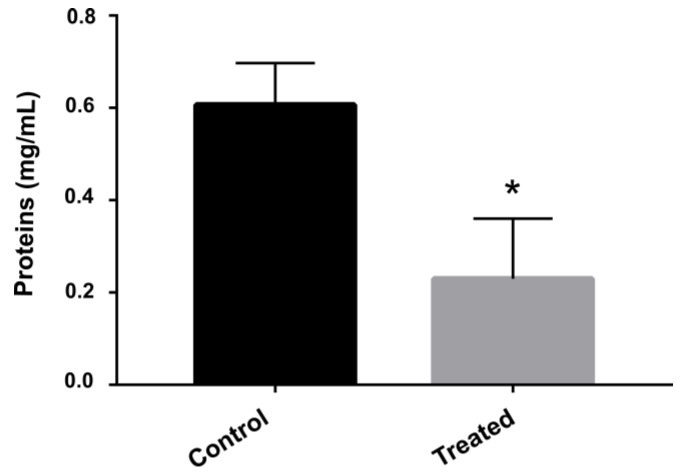
**Protein quantification after exposure to AOS.** Protein concentration was measured after exposure of *P. brasiliensis* for 12 h to AOS. The * indicates a significant difference between the samples evaluated at that point with *p* ≤ 0.05. Error bars represent the standard deviation of three biological replicates.

**Figure 4 jof-09-00066-f004:**
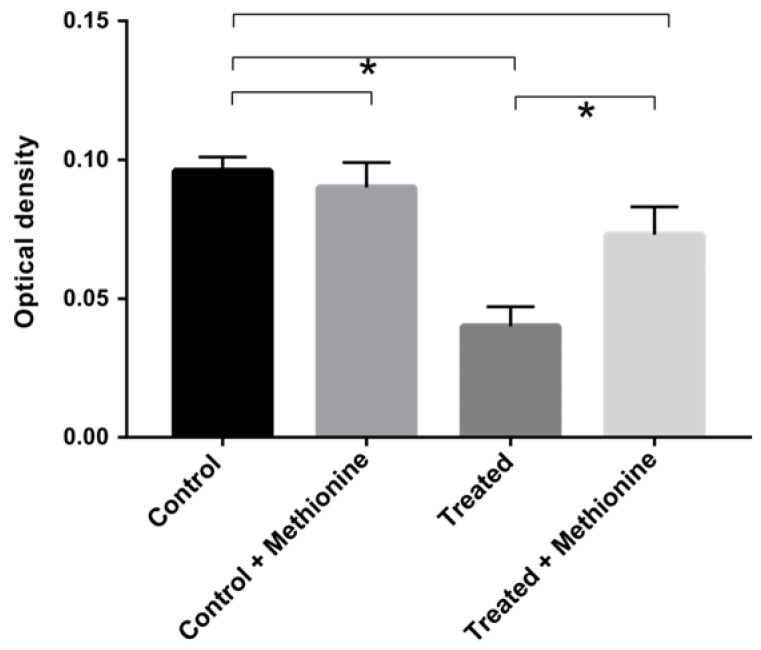
**Growth curve of *P. brasiliensis* exposed to A0SFF in medium supplemented with methionine.** The restoration of growth of *P. brasiliensis* cultured in the absence (control) and presence (treated) of AOS for 12 h was verified after supplementation of the culture medium with methionine. * Means significant difference between the samples (*p* ≤ 0.05). Error bars represent the standard deviation of three biological replicates.

**Figure 5 jof-09-00066-f005:**
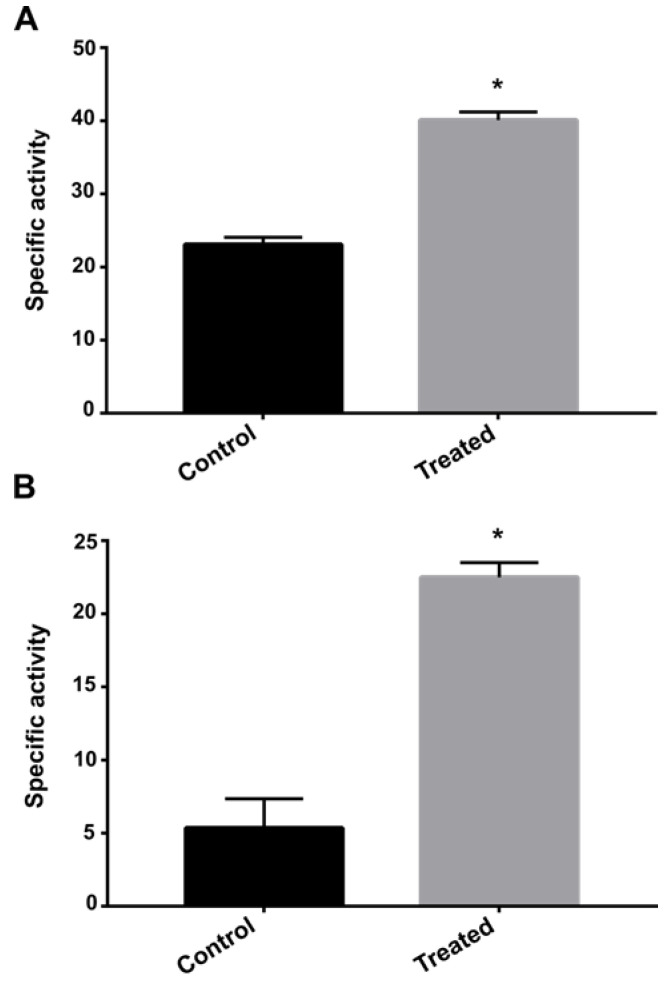
**Enzymatic activity of formamidase and urease after exposure to AOS.** Enzyme activity was verified after 12 h of exposure to the compound. (A) Formamidase activity. (B) Urease activity. Error bars represent the standard deviation of three biological replicates. The statistically significant difference between the control and treated samples at each time was verified using Student’s *t*-test. The * means a significant difference (*p* ≤ 0.05) between samples.

**Figure 6 jof-09-00066-f006:**
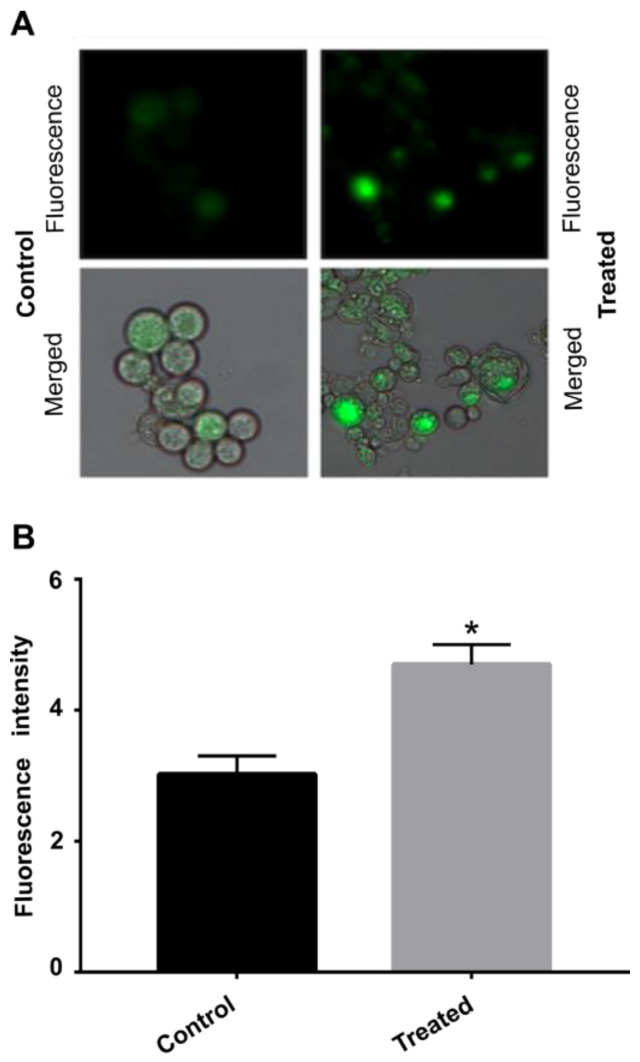
**Production of ROS by *P. brasiliensis* after exposure to AOS.** (**A**) Fluorescence of *P. brasiliensis* cells exposed or not to AOS for 12 h, visualized by fluorescence microscopy after staining with DCFH-DC. (**B**) Fluorescence intensity in pixels. Error bars represent the standard deviation of three biological replicates. The statistically significant difference between the control and treated samples at each time was verified using Student’s *t*-test. The * means a significant difference (*p* ≤ 0.05) between samples.

**Figure 7 jof-09-00066-f007:**
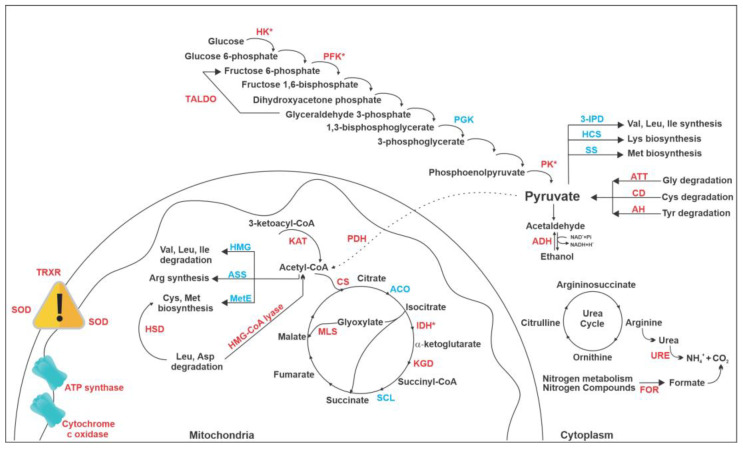
**AOS mode of action according to the proteomic profile of *P. brasiliensis* treated with the compound.** The compound AOS differentially regulated proteins related to glycolysis, TCA, glyoxylate cycle, urea cycle and amino acid metabolism, including homoserine dehydrogenase (HSD). Regulatory enzymes, such as pyruvate kinase (PK), 6-phosphofructokinase (PFK) and isocitrate dehydrogenase (IDH) are upregulated, showing that pyruvate production and TCA is stimulated but the electron transport chain and ATP synthesis enzymes are downregulated, reducing the ability of cells treated with AOS to produce energy and consequently grow. Hexokinase (HK); Phosphoglycerate kinase (PGK); 3-isopropylmalate dehydratase (3-IPD); Homocysteine S-methyltransferase (HCS ); Saccharopine dehydrogenase (SS); Amino methyltransferase (ATT); Cysteine dioxygenase (CD); Acylpyruvate hydrolase (AH); Aldehyde dehydrogenase (ADH); Pyruvate dehydrogenase (PDH); Aconitase (ACO); α-ketoglutarato dehydrogenase (KGD); Succinate-CoA ligase (SCL); Malate synthase (MLS); 3-ketoacyl-CoA thiolase B (KAT); Citrate synthase (CS); 3-hydroxy-3-methylglutaryl coenzyme A synthase (HMG); 5-methyltetrahydropteroyltriglutamate-homocysteine S-methyltransferase (MetE); Arginosuccinate synthase (ASS); Hydroxymethylglutaryl-CoA lyase (HMG-CoA lyase); Superoxide dismutase (SOD); Thioredoxin reductase (TRXR); urease (URE ) formamidase (FOR). Red indicates upregulated proteins and blue indicates downregulated proteins.

**Table 1 jof-09-00066-t001:** Antifungal and cytotoxic activity of the AOS compound.

Isolate	MIC	MFC	CC	SI
*P*01	6.09/1.95	6.09/1.95	4685.88/1500	769.43
*P*03	0.75/0.24	0.75/0.24		6247.84
*Pb*18	0.75/0.24	0.75/0.24		6247.84
EPM83	1.5/0.48	1.5/0.48		3123.92

MIC-Minimum Inhibitory Concentration; MFC-Minimum Fungicidal Concentration; CC-Cytotoxic Concentration; SI-Selectivity Index. *P. lutzii* (*P*01); *P. brasiliensis* (*Pb*18); *P. americana* (*P*03); *P. restrepiensis* (EPM83). MIC, MFC and CC are expressed µM/µg.mL^−1^). Molecular weight of AOS: 320.11.

## Data Availability

The submission of the proteomics’ results was assigned the identifier PASS03787 in the PeptideAtlas repository. https://db.systemsbiology.net/sbeams/cgi/PeptideAtlas/PASS_View?identifier=PASS03787.

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
