# Peer review of "Proteomic Response of Paracoccidioides brasiliensis Exposed to the Antifungal 4-Methoxynaphthalene-N-acylhydrazone Reveals Alteration in Metabolism"

_jof, 2022, doi:10.3390/jof9010066_

Round 1

Reviewer 1 Report

The manuscript describes the evaluation of a novel antifungal (designated AOS) that targets the homoserine dehydrogenase (HSD) in Paracoccidioides sp that is needed for the synthesis of methionine, threonine and lysine in fungi and plants.  The investigators initially demonstrate inhibitory and antifungal activities in various isolates and compared to cytotoxicity BALB/3T3 cell with as much as at >6000 selectivity index against P. brasiliensis compared to cell cytotoxicity.  They include data for MIC and MFC which are identical in each case so it is not clear how they define/distinguish each.  Nonetheless they show some supportive data that the inhibition does target HSD by including a supplementation experiment with methionine that rescues the inhibition of growth in the presence of the AOS inhibitor.  The remain experimentation is directed at profiling and validating protein changes or projected biological processes effected by the inhibitor based on proteomics profiling, where they report a large number of proteins that were up or down regulated in the treated versus control groups.  Some concerns about the experimental design which may have a negative impact on the data interpretation include:

1.    Most proteins in each group, as reported in Table S1, were detected in only the Treated group (~75%) or only in the control group (~50%) which is a bit uncommon for a whole lysate proteomics experiment where one typically sees modulation of proteins changes rather than all or none changes.

2.    Data collection appears to be as 3 replicate injection of the same sample with no p values for the significance of the changes reported in Table S1 which makes it challenging to evaluate whether the comparisons are valid between the two groups.

3.    If the goal is to understand the changes associated with the mechanism of the drug treatment, evaluation of an earlier timepoint may be warranted.  While 12 hours seems like a reasonable choice given that they maintain 85% viability at that stage, but given that by 24hr, viability is down to 28% would indicate that the fungi were likely well down the path toward cell death even by 12 hr.  As such even at 12 hr the protein changes may not be related to the drug targeting or mechanism of action but may be more related the consequences of reduced cell viability (i.e. general downstream proteins associated with dying cells independent of the drug mechanism).  This coupled with the lack of statistical significance data potentially rendering the subsequent data interpretation as less informative about the specific role of the inhibitor and more descriptive. 

On the positive side the investigators do follow up with some additional enzyme studies and ROS activation measurements to support some of the observation that they see and the hypotheses that they derive from to differential protein profiling. 

Outside for some of the experimental design concerns mentioned above, the technical approach to sample preparation and proteomics data collection and subsequence validation assay all appear to have been done in a sound manner.  Thus overall the manuscript does provide some additional insight into a potential new therapeutic that targets a non-mammalian enzyme target and thus that in itself may be significant, but increased rigor and reproducibility of the profiling with multiple biological replicates would benefit this study as would evaluating an earlier timepoint before the significant drops in viability.

Author Response

Thank you for taking your time to evaluate our manuscript. The comments were very helpful and constructive and allowed us to significantly improve the manuscript. We expect that the responses below have helped to clarify the potential problems raised about our work.

Comments

If one of the referees has suggested that your manuscript should undergo
extensive English revisions, please address this issue during revision. We
propose that you use one of the editing services listed at
https://www.mdpi.com/authors/english or have your manuscript checked by a
native English-speaking colleague.

Comments and Suggestions for Authors

The manuscript describes the evaluation of a novel antifungal (designated AOS) that targets the homoserine dehydrogenase (HSD) in Paracoccidioides sp that is needed for the synthesis of methionine, threonine and lysine in fungi and plants.  The investigators initially demonstrate inhibitory and antifungal activities in various isolates and compared to cytotoxicity BALB/3T3 cell with as much as at >6000 selectivity index against P. brasiliensis compared to cell cytotoxicity.  They include data for MIC and MFC which are identical in each case so it is not clear how they define/distinguish each.  Nonetheless they show some supportive data that the inhibition does target HSD by including a supplementation experiment with methionine that rescues the inhibition of growth in the presence of the AOS inhibitor.  The remain experimentation is directed at profiling and validating protein changes or projected biological processes effected by the inhibitor based on proteomics profiling, where they report a large number of proteins that were up or down regulated in the treated versus control groups.  Some concerns about the experimental design which may have a negative impact on the data interpretation include:

  1. Most proteins in each group, as reported in Table S1, were detected in only the Treated group (~75%) or only in the control group (~50%) which is a bit uncommon for a whole lysate proteomics experiment where one typically sees modulation of proteins changes rather than all or none changes.

Response: Our group works with P. brasiliensis proteomes that involve its response to antifungal compounds, these results seem to be standard in these cases [1,2,3]. One possibility is that the fungus is really going through a stress that leads to its death, in this way it inhibits proteins that used to be natively expressed, that's why they only appear in the control, and it produces a response activating other enzymes that were inert, that's why it only appears in the treatment condition.

  1. Rocha, O.B.; do Carmo Silva, L.; de Carvalho Júnior, M.A.B.; de Oliveira, A.A.; de Almeida Soares, C.M.; Pereira, M. In Vitro and in Silico Analysis Reveals Antifungal Activity and Potential Targets of Curcumin on Paracoccidioides Spp. Braz J Microbiol 2021, 52, 1897–1911, doi:10.1007/s42770-021-00548-6.
  2. e Silva, K.S.; da S Neto, B.R.; Zambuzzi-Carvalho, P.F.; de Oliveira, C.M.; Pires, L.B.; Kato, L.; Bailão, A.M.; Parente-Rocha, J.A.; Hernández, O.; Ochoa, J.G.; et al. Response of Paracoccidioides Lutzii to the Antifungal Camphene Thiosemicarbazide Determined by Proteomic Analysis. Future Microbiology 2018, 13, 1473–1496, doi:10.2217/fmb-2018-0176.
  3. Prado, R.S.; Bailão, A.M.; Silva, L.C.; de Oliveira, C.M.A.; Marques, M.F.; Silva, L.P.; Silveira-Lacerda, E.P.; Lima, A.P.; Soares, C.M.; Pereira, M. Proteomic Profile Response of Paracoccidioides Lutzii to the Antifungal Argentilactone. Frontiers in Microbiology 2015, 6, doi:10.3389/fmicb.2015.00616.

  1. Data collection appears to be as 3 replicate injection of the same sample with no p values for the significance of the changes reported in Table S1 which makes it challenging to evaluate whether the comparisons are valid between the two groups.

Response: It was performed biological replicates in equimolar quantities of the control and the MCS inhibitor compound treatment. Then the samples of each treatment were polled and then for each sample were analyzed in three technical replicates. All statistical analysis were performed by ExpressionE software v3.0 package (Waters, UK). The ion detection spectra counting, clustering and log-scale parametric normalization procedures were performed into PLGS with ExpressionE license installed. The calculation of the log ratio and the confidence interval was based on a Gaussian distribution model, which allows for the possibility of an uncertain peptide assignment, an incorrect assignment of data to a cluster or an interference. The confidence interval of 95% was used. This statistical model and data processing parameters has been used to perform proteomics studies. Below some examples are listed:

Rocha, O.B.; do Carmo Silva, L.; de Carvalho Júnior, M.A.B.; de Oliveira, A.A.; de Almeida Soares, C.M.; Pereira, M. In Vitro and in Silico Analysis Reveals Antifungal Activity and Potential Targets of Curcumin on Paracoccidioides Spp. Braz J Microbiol 2021, 52, 1897–1911, doi:10.1007/s42770-021-00548-6.

Prado, R.S.; Bailão, A.M.; Silva, L.C.; de Oliveira, C.M.A.; Marques, M.F.; Silva, L.P.; Silveira-Lacerda, E.P.; Lima, A.P.; Soares, C.M.; Pereira, M. Proteomic Profile Response of Paracoccidioides Lutzii to the Antifungal Argentilactone. Frontiers in Microbiology 2015, 6, doi:10.3389/fmicb.2015.00616.

Silva, W. M., R. D. Carvalho, S. C. Soares, I. F. Bastos, E. L. Folador, G. H. Souza, Y. Le Loir, A. Miyoshi, A. Silva and V. Azevedo (2014). "Label-free proteomic analysis to confirm the predicted proteome of Corynebacterium pseudotuberculosis under nitrosative stress mediated by nitric oxide." BMC Genomics 15: 1065.

Levin Y, Hradetzky E, Bahn S. Quantification of proteins using data-independent analysis (MSE) in simple andcomplex samples: a systematic evaluation. Proteomics. 2011 Aug;11(16):3273-87.

da Cunha, N. B., A. M. Murad, G. R. Vianna, C. Coelho and E. L. Rech (2013). "Expression and characterization of recombinant molecules in transgenic soybean." Curr Pharm Des 19(31): 5553-5563.

Cunha, N. B., A. M. Murad, G. L. Ramos, A. Q. Maranhao, M. M. Brigido, A. C. Araujo, C. Lacorte, F. J. Aragao, D. T. Covas, A. M. Fontes, G. H. Souza, G. R. Vianna and E. L. Rech (2011). "Accumulation of functional recombinant human coagulation factor IX in transgenic soybean seeds." Transgenic Res 20(4): 841-855.

da Costa, M. R., L. Pizzatti, R. S. Lindoso, J. F. Sant'Anna, B. DuRocher, E. Abdelhay and A. Vieyra (2014). "Mechanisms of kidney repair by human mesenchymal stromal cells after ischemia: a comprehensive view using label-free MS(E)." Proteomics 14(12): 1480-1493.

Panis, C., L. Pizzatti, A. C. Herrera, S. Correa, R. Binato and E. Abdelhay (2014). "Label-free proteomic analysis of breast cancer molecular subtypes." J Proteome Res 13(11): 4752-4772.

  1. If the goal is to understand the changes associated with the mechanism of the drug treatment, evaluation of an earlier timepoint may be warranted.  While 12 hours seems like a reasonable choice given that they maintain 85% viability at that stage, but given that by 24hr, viability is down to 28% would indicate that the fungi were likely well down the path toward cell death even by 12 hr.  As such even at 12 hr the protein changes may not be related to the drug targeting or mechanism of action but may be more related the consequences of reduced cell viability (i.e. general downstream proteins associated with dying cells independent of the drug mechanism).  This coupled with the lack of statistical significance data potentially rendering the subsequent data interpretation as less informative about the specific role of the inhibitor and more descriptive. 

On the positive side the investigators do follow up with some additional enzyme studies and ROS activation measurements to support some of the observation that they see and the hypotheses that they derive from to differential protein profiling. 

Outside for some of the experimental design concerns mentioned above, the technical approach to sample preparation and proteomics data collection and subsequence validation assay all appear to have been done in a sound manner.  Thus overall the manuscript does provide some additional insight into a potential new therapeutic that targets a non-mammalian enzyme target and thus that in itself may be significant, but increased rigor and reproducibility of the profiling with multiple biological replicates would benefit this study as would evaluating an earlier timepoint before the significant drops in viability.

Response: Our group have established a standard to perform this analysis at a time point where more than 90% of the cells would be viable [1,2] in order to verify the response to compound prior to cell death. Using this criterion, we have successfully validated the proteomic data, which also occurred here. It is noteworthy that for all-time points analyzed (9, 12, and 24 hours), the results corroborated among them and with proteomic data. Additionally, proteomic analysis is a process that takes a very long time to complete.

  1. Rocha, O.B.; do Carmo Silva, L.; de Carvalho Júnior, M.A.B.; de Oliveira, A.A.; de Almeida Soares, C.M.; Pereira, M. In Vitro and in Silico Analysis Reveals Antifungal Activity and Potential Targets of Curcumin on Paracoccidioides Spp. Braz J Microbiol 2021, 52, 1897–1911, doi:10.1007/s42770-021-00548-6.
  2. Prado, R.S.; Bailão, A.M.; Silva, L.C.; de Oliveira, C.M.A.; Marques, M.F.; Silva, L.P.; Silveira-Lacerda, E.P.; Lima, A.P.; Soares, C.M.; Pereira, M. Proteomic Profile Response of Paracoccidioides Lutzii to the Antifungal Argentilactone. Frontiers in Microbiology 2015, 6, doi:10.3389/fmicb.2015.00616.

Reviewer 2 Report

1. Line 77 of materials and methods - "com" is placed after "according". Should this be changed to "according to"?

2. Line 97 of materials and methods - Please the first minimum fungicidal to minimum inhibitory since this is describing the MIC and not the MFC.

3. Line 174 of material and methods - What do the authors mean by "Posteriorly"?

4. Results all units for MIC, MFC, CC, and SI are provided in uM.  Please also include mg/L (or mcg/mL) below these values and in parenthesis.

5. Is Figure 7 original, or was it derived from other sources?  If derived from sources other than the results of this work, please also provide citations.

Author Response

Thank you for taking your time to evaluate our manuscript. The comments were very helpful and constructive and allowed us to significantly improve the manuscript. We expect that the responses below have helped to clarify the potential problems raised about our work. 

Comments and Suggestions for Authors

  1. Line 77 of materials and methods - "com" is placed after "according". Should this be changed to "according to"?

Response: The requested change has been made.

  1. Line 97 of materials and methods - Please the first minimum fungicidal to minimum inhibitory since this is describing the MIC and not the MFC.

Response: The requested change has been made.

  1. Line 174 of material and methods - What do the authors mean by "Posteriorly"?

Response: The phrase “Posteriorly, the material was centrifuged for 20 min at 5000 g and the supernatant was collected was changed to “The lysate was centrifuged for 20 min at 5000 g and the supernatant was collected”.

  1. Results all units for MIC, MFC, CC, and SI are provided in uM.  Please also include mg/L (or mcg/mL) below these values and in parenthesis.

Response: The results of MIC, MFC and CC were provided in µM and µg/mL.

  1. Is Figure 7 original, or was it derived from other sources?  If derived from sources other than the results of this work, please also provide citations.

Response: The figure 7 is original and represents a summary of the main modulation of proteomic profile of P. brasiliensis treated with the AOS.

Round 2

Reviewer 1 Report

The authors provide a rebuttal by do not appear to have made any additional clarification in the revised manuscript.  Specifically, even with the additional explanation on the rationale behind the approach to pool multiple samples and then divide them out as technical replicates to evaluate statistical significance, it is not clear how this approach provides any measure of biological significance.  Why not just run each of the biological replicates independently and then one can evaluate the biological significance as well.  With the current approach, all that is being measured is the technical variability of the sample preparation and mass spectrometry which has little to do with biological reproducibility/significance of the drug response instead all that is measure in and average N=1 experiment.   At the very least, the methods section should be very clear about the number of samples pooled in each control v treatment group and then how the samples were split out into multiple injections so that the reader understands that the statistics are based on technical replicates.  In the end, if technical replicates are acceptable to the editors and the journal, then this reviewer agrees to move forward with publication provided that the details are spelled out more clearly in the methods section of the manuscript.